# Cognition and Functionality Were Not Affected Due to the COVID-19 Pandemic in People with Mild Cognitive Impairment and AD Dementia Attending Digital Non-Pharmacologic Interventions

**DOI:** 10.3390/brainsci13071044

**Published:** 2023-07-08

**Authors:** Marianna Tsatali, Despina Moraitou, Evgenia Sakka Boza, Magdalini Tsolaki

**Affiliations:** 1Greek Association of Alzheimer’s Disease and Related Disorders, 54643 Thessaloniki, Greece; eugeniasampo@gmail.com (E.S.B.); tsolakim1@gmail.com (M.T.); 2Laboratory of Psychology, Department of Cognitive and Experimental Psychology, Faculty of Philosophy, School of Psychology, Aristotle University of Thessaloniki (AUTh), 54124 Thessaloniki, Greece; despinamorait@gmail.com; 3Lab of Neurodegenerative Diseases, Center for Interdisciplinary Research and Innovation, Aristotle University of Thessaloniki (CIRI—AUTh), 54124 Thessaloniki, Greece; 41st Department of Neurology, Medical School, Aristotle University of Thessaloniki (AUTh), 54124 Thessaloniki, Greece

**Keywords:** COVID-19 pandemic effect, Alzheimer’s disease dementia, mild cognitive impairment, COVID-19 quarantine

## Abstract

Background: The majority of previous studies showed that older adults with mild cognitive impairment (MCI) as well as Alzheimer’s disease dementia (ADD) had impaired cognition and mood status, as well as increased behavioral disturbances after the first wave of the COVID-19 pandemic. However, there are still controversial data as regards the multifactorial impact of the restrictive measures on cognition, mood and daily function in older adults with MCI and ADD. Aim: In the current study, the scope is to identify possible deterioration by means of cognitive and functional level due to mood and behavioral alterations during the second quarantine imposed in Greece between November 2020 and May 2021, as well as one year after the second quarantine, in May 2022. Methods: Participants were recruited from the two day centers of the Greek Association of Alzheimer Disease and Related Disorders (GAADRD). They underwent three yearly follow up assessments from May 2020 to May 2022 and participated in cognitive training interventions (through digital online means) during the aforementioned period. Mixed measures analyses of variance as well as path models were used for the study’s purposes. Results: The study sample comprised 210 participants (175 people with MCI and 35 people with ADD). The mean age was 71.59 and 77.94 for people with MCI and mild ADD, respectively, whereas the average number of years of education was 12.65 for those with MCI and 9.83 for people with mild ADD. The results show that participants’ deterioration rate (D), calculated by means of their performance in neuropsychological and functional assessments between 2020 and 2021 (D1) and 2021 and 2022 (D2), did not change significantly, except for the Rey Auditory Verbal Learning Test (RAVLT), since both groups displayed a larger D2 across the test conditions (immediate recall, fifth trial and delayed recall). Trail Making Test-B (TMT-B) performance, applied only in the MCI group, decreased more in relation to the deterioration rate D2. Additionally, two path models were applied to measure the direct relationships between diagnosis, performance in tests measuring mood and neuropsychiatric disturbances (NPI) and cognition, as measured by the RAVLT, in the 2020–2022 assessments. TMT-B was administered only in the MCI population, and therefore was not included in path models. The results show that participants’ scores in RAVLT conditions were related to diagnosis and NPI performance, which was positively affected by diagnosis. No other relationships between RAVLT with mood tests were observed. Conclusions: Our results show that after the second lockdown period, the neuropsychological performance of people with MCI and ADD, calculated by means of their D2, did not change, except from their verbal memory, as well as visual scanning and information processing, measured using the TMT-B. Therefore, it can be assumed that those who were enrolled in digital non-pharmacological interventions during the COVID-19 pandemic home restrictions did not experience increased cognitive and functional deterioration due to mood and behavioral alterations after the pandemic.

## 1. Introduction

The COVID-19 pandemic affected the entire planet, due to the restrictions imposed to protect public health. Both the pandemic, as well as the restrictions posed, have been devastating for the vulnerable groups of the population, including older adults [1]. In addition to the high mortality, studies [2] have shown that older adults faced additional physical and psychological problems due to the established situation.

Research conducted with patients with mild cognitive impairment (MCI), Alzheimer disease dementia (ADD) and other dementias showed that the quarantine led to increased rates of anxiety, depression, sleep disturbances, irritability and eating disturbances [3,4,5]. Interestingly, some researchers interpreted this increase due to increased sleep disturbances and irritability, as a result of social isolation and disruption of daily life habits and leisure activities [6]. Increased levels of anxiety and depression (including mood swings), compared to the pre-pandemic period, were observed in patients with ADD who lived in retirement homes, due to the reduction in social activities and interactions, even with their roommates [7,8]. Other problems that appeared in patients during the quarantine were delusions and delirium [9], whereas the most frequent emotions reported by patients were anger, fear and loneliness, mainly concerning people who lived alone and could not be visited by family members. Regarding the physical condition of the patients, a decrease was observed in their energy levels and their involvement in daily activities, including either outdoor activities or household tasks [10,11]. On the other hand, some other studies, in contrast to the aforementioned findings, showed no statistically significant increase in depression, nor in appetite loss or sleep quality deterioration [10,12]. In the study by Boutoleau-Bretonnière et al. [13], a portion (10 out of 38) of people with ADD and the behavioral variant of frontotemporal dementia (bvFTD) manifested more intense neuropsychiatric symptoms. It is worth noting that in people with ADD, increased agitation was more common than in those with bvFTD.

The outbreak of the pandemic and the subsequent quarantine, in addition to having psychological effects in people living with dementia (PwD), affected their general functionality. People with mild and moderate neurocognitive disorders suffered significant decline in their ability to feed, get dressed, verbally communicate, and socialize [14].

The pandemic and its effects on cognition have also been studied with regard to patients with dementia. Studies found that during the pandemic, there was a larger decline in patients’ scores in the Mini-Mental State Examination (MMSE) compared to the period prior to quarantine [15]. Of particular interest is the fact that patients with mild dementia showed a greater decline in their cognitive abilities compared to patients with more severe levels of dementia [16]. More specifically, the cognitive areas in which dementia patients experienced a statistically significant decline in their performance were memory, calculation, orientation, drawing designs and language. These findings were attributed to isolation and a lack of social interactions during the quarantine [15]. Furthermore, an unexpected finding was derived in a survey of older adults with MCI in Argentina. Specifically, participants who were living alone performed better at cognitive ability tests than participants who were living with others. Moreover, the researchers underlined the importance of cognitive–behavioral training programs as a protective factor, given that the participants in their research attended this type of program with health professionals [17].

Our previous work (Tsatali et al., 2021) [18] in people with MCI and ADD presented contradictory findings compared to the results in the existing literature. Specifically, people who followed remote, non-pharmacological digital cognitive training programs during the period of the first lockdown in Greece did not experience cognitive, mood, functional and behavioral deterioration compared to the pre-pandemic period. A notable work on the effects of the pandemic on Alzheimer’s patients is that of Gan et al. [9] in China. In this study, the researchers recruited a group of participants with MCI and dementia who participated in their assessment a year before the COVID-19 pandemic (control group). Afterwards, they compared these records to the assessments of people with MCI and dementia performed made during the pandemic (experimental group). Νο statistical changes in cognitive evaluation or the evaluation of neuropsychiatric symptoms was observed, except for increased sleep disturbances.

Conclusively, it seems that the pandemic had affected the cognitive, psychological and behavioral levels of patients with dementia; however, this relationship is not direct, whereas significant variables, such as psychological support [19,20,21], attending cognitive training programs, and social isolation should be also taken into account. The purpose of this research is to investigate the cognitive and psychological effects of the pandemic on patients with MCI and ADD during the second period of quarantine in Greece, and therefore to identify whether the second pandemic wave had the same effect in these participants just after the end of home restrictions in 2021, as well as one year after, in 2022.

### Aim of the Study

The hypothesis of the current study is that (1a) people with MCI and ADD did not experience cognitive and functional impairment (1b) as a result of emotional and behavioral changes during the second lockdown in Greece—the time period after the first quarantine in 2020 and the second one imposed in 2021 (November 2020 to the middle of May 2021)—as well as one year after the end of the second wave of the COVID-19 pandemic.

## 2. Methods

### 2.1. Participants

Participants were recruited from the day care centers of “Saint John” and ‘’Saint Helen” of the Greek Association of Alzheimer’s Disease and Related Disorders (GAADRD) in Thessaloniki. The sample consisted of the following groups: (a) people with MCI (amnestic and non-amnestic) and (b) people living with ADD according to the Diagnostic and Statistical Manual of Mental Disorders, fifth edition (DSM-5) [22]. Specifically, after receiving the diagnosis in 2018, they attended onsite non-pharmacological interventions in 2019 and 2020 (before the pandemic period). After the first wave of the COVID-19 pandemic, they attended digital cognitive interventions until the end of the second wave of the pandemic. Participants underwent the follow-up evaluations during the same three-month period (May–July) in 2020, 2021 and 2022 to identify whether the imposition of the second quarantine further affected their cognitive and functional abilities due to mood and behavioral disturbances attributed to the restrictive measures related to the pandemic.

In more detail, participants followed the diagnostic protocol of the GAADRD, which involves neuropsychiatric and neuropsychological assessment, neurological evaluation, neuroimaging, as well as blood tests to exclude other types of reversible cognitive impairment or dementia. The official neuropsychological assessment was administered by the neuropsychologists who constituted the expert group of the GAADRD. More details regarding the initial diagnostic process, as well as the online and onsite non-pharmacological programs which our participants attended during the first and second quarantine periods, are included in our previous paper [21].

The inclusion criteria for MCI were in line with the DSM-5 criteria [22]: (a) Mini Mental State Examination (MMSE) total score ≥ 26, (b) stage 3 of the disease according to the Global Deterioration Scale [23], and (c) 1.5 standard deviations (SD) below the normal mean in at least one cognitive domain according to the neuropsychological tests administered. The inclusion criteria for ADD were: (a) diagnosis of Alzheimer’s disease according to the DSM-5, (b) MMSE total score ≤ 23 and ≥12, (c) stage 4 and 5 of the disease according to the Global Deterioration Scale, and (d) 2 standard deviations (SD) below the normal mean according to age and education in at least one cognitive domain according to the utilized neuropsychological tests as well as subsequent deterioration in independent living. The exclusion criteria for both groups were as follows: (a) a history of psychiatric illness, (b) substance abuse or alcoholism, (c) history of traumatic brain injury, (d) neurological disorders, (e) thyroid disorders, (f) diabetes, (g) drug treatment with opioids, and (h) severe sensory deficits. An additional exclusion criterion for the older adults with ADD was those who had ADD at a severe stage.

### 2.2. Procedure

The neuropsychological evaluations administered in the last two follow-up assessments just after the first COVID 19 pandemic lasted for approximately two hours in two different onsite sessions. At first, the screening protocol, adapted into Greek, included the following tests: Mini Mental State Examination (Greek cut-off scores from Fountoulakis et al. [24]), and Montreal Cognitive Assessment (Greek cut-off scores from Poptsi et al. [25]). Short- and long-term as well as verbal memory/learning were measured using the Rey’s Verbal Learning Test (three variables; RAVLT first trial measures short-term memory, RAVLT fifth trial measures verbal learning, and RAVLT delayed recall measures long-term memory; Greek cut-off scores from Messinis et al. [26]). Verbal fluency was measured using the phonemic verbal fluency test adapted into Greek by Kosmides et al. [27]. Visuospatial function was measured using the RCFT copy as well as delayed recall trials, adapted by Tsatali et al. [28,29]. Finally, the Trail Making test (TMT)-part B (adapted into Greek by Zalonis et al. [30]) subtest was administered to measure processing speed. The Functional Cognitive Assessment Scale (Greek cut-off scores from Kounti et al. [31]) and Functional Rating Scale for Symptoms of Dementia (FRSSD [32]) were used to measure independent living. The Short Anxiety Screening Test (SAST) [33] as well as Geriatric Depression Scale (GDS) [34] were used in order to measure mood deficits. Finally, the Greek Neuropsychiatric Inventory (NPI) [35] was implemented with a family member of participants to measure the level of neuropsychiatric symptoms. In more detail, NPI measures neuropsychiatric disturbances.

### 2.3. Ethics

Similar to the initial assessment as well as the two follow-up measurements, participants read and signed the informed consent before each consecutive neuropsychological evaluation in 2021–2022. A legal representative, typically the main caregiver of people with ADD in the mild and moderate stages, read the consent form and signed the relevant consensus documents.

The study was approved by the Scientific and Ethics Committee of the GAADRD (Scientific Committee Approved Meeting Number: 88/4-5-2023), which follows the new General Data Protection Regulation (EU) 2016/679 of the European Parliament and of the Council of 27 April 2016 on the protection of natural persons with regard to the processing of personal data and on the free transfer of such data, as well as the principles outlined in the Helsinki Declaration.

### 2.4. Data Analysis

The SPSS software version 28 (IBM; SPSS Statistics for Windows, Version 28.0. Armonk, NY: IBM Corp 28.0) was initially used to perform the statistical analyses. In order to examine whether there were any significant differences between the groups and their deterioration rates (related to the assessment times from after the first quarantine imposed in spring 2020 to one year after the end of the second wave of the pandemic), mixed-measure ANOVA (2 groups: people with MCI, people with ADD) × 2 (deterioration rate—D: D1 = deterioration difference between 2020 and 2021 assessments; D2 = deterioration difference between 2021 and 2022 assessments) was used. The D1 deterioration rate was measured by subtracting participants’ records in 2021 from those in 2020, and the D2 deterioration rate was the result of the subtraction between participants’ performance in 2021 from those in 2022.

Hence, the higher the D indices, the higher the deterioration of performance as regards the tests in which the higher the sum scores, the higher the participant’s performance. Vice versa, the higher the D indices, the lower the deterioration of participants’ performance for the tests in which higher scores indicate worse performance, e.g., NPI, GDS, FRSSD, FUCAS and TMT B, where a higher score describes worsening performance and, therefore, the lower the D indices, the greater the deterioration. MoCA, Digit forward and backward as well as TMT B were administered only in those with MCI.

In order to examine the direction of the relationships between mood and cognitive variables in two neuropsychological assessments (2021 and 2022), path analysis was conducted in EQS (version 6.4) statistical software [36]. A maximum likelihood estimation procedure was performed. Regarding the confirmation of a path model, a non-significant level of the Goodness of Fit index χ^2^, namely *p* > 0.05, is indicative of a good fit of the model to the data. However, this index is affected by the sample size. Meanwhile, when the value of the root mean square error of approximation (RMSEA) is <0.05, it is an indication of the good fit of the model to the data. RMSEA values ranging from 0.06 to 0.08 indicate a reasonable and, therefore, acceptable approximation error. Additional support for the fit of the solution is evidenced by a 90% confidence interval range of the RMSEA whose upper limit is below the cut-off values of 0.08–0.10. However, the RMSEA value is relatively “expanded” in cases of a small sample size, and this is reflected in the confidence interval range. This means that RMSEA should be considered as a model fit index, but with caution and taking account other indices as well [37]. The SRMR index is an absolute measure of fit and is defined as the standardized difference between the observed correlation and the predicted correlation. It is a positively biased measure, and this bias is greater for small-N and for low-df studies. Because the SRMR is an absolute measure of fit, a value of zero indicates perfect fit. A value less than 0.08 is generally considered a good fit. Conversely, the Comparative Fit Index (CFI) examines whether the data fit a hypothesized path model compared to the basic model. Values greater than 0.90 indicate adequate fit of the model to the data, whereas values close to 1.00 indicate a good fit [38]. To improve model fit, we examined the modification indices, namely the Wald and the Lagrange tests, which represent frequently used statistics to identify focal areas of a misfit in a path analysis solution [38].

## 3. Results

### 3.1. Descriptive Statistics

Overall, 210 participants took part in the current study, including (a) 175 older adults with MCI (59 men and 116 women, age range: 51 to 92 years, M = 71.59, SD = 6.93, education range: 2 to 20 years, M = 12.65, SD = 3.74) and (b) 35 older adults with ADD in mild and moderate stages (17 men and 18 women, age range: 63 to 90 years, M = 77.94, SD = 6.49, education range: 2 to 18 years, M = 9.83, SD = 5.14). It must be mentioned that approximately 90% of participants had received a laboratory-confirmed diagnosis of COVID 19 after being asked by the doctor of the GAADRD. No significant differences were found between the two groups according to gender (χ(1) = 2.788, *p* = 0.095. On the contrary, significant differences were observed between the two groups regarding demographic characteristics such as age (F(1, 208) = 0.033, *p* < 0.001) and education (in years) (F(1, 208) = 41.63, *p* < 0.001). As expected, older adults with ADD were older and less educated. The sample’s demographic data are presented in Table 1.

Raw scores, namely mean and standard deviation values, from the three assessments between 2020 and 2022, as well as their *p* scores are presented in Table 2.

### 3.2. The Effects of Diagnostic Group and Deterioration Rate on Cognition, Daily Functioning, Mood and Behavior

#### 3.2.1. General Cognitive Status (MMSE, MoCA)

As regards the MMSE deterioration rates, higher D1 and D2 values indicate lower levels of general cognitive status. Repeated measures analysis revealed a significant main effect of the diagnostic group, which means that MCI participants had a lower deterioration rate in MMSE scores (M = 0.186, SE = 0.101) as compared to participants with ADD (M = 1.632, SE = 0.225), as expected. Concerning the MoCA test performance, no significant differences were found in people with MCI performance between 2020 and 2022.

#### 3.2.2. Activities of Daily Living—Everyday Functioning (FUCAS)-Functional Rating Scale of Symptoms of Dementia (FRSSD)

High FUCAS and FRSSD test scores indicate low performance, and therefore the higher the D rate’s mean value, the better the level of daily functioning. By means of the FUCAS test, the results show a significant main effect of the group—that is, MCI participants had a lower deterioration rate (M = −0.401, SE = 0.24) than the ADD group (M = −4.19, SE = 0.53), again as expected. The interaction between group and condition was significant. The subsequent Scheffe post hoc test showed that the D2 value was higher in the ADD group as compared to MCI participants (I–J = 3.796, *p* < 0.001). The same results were found for FRSSD, which indicates a significant main effect of the group—that is, MCI participants had a lower deterioration rate (M = −0.184, SE = 0.13) than the ADD group (M = −2.16, SE = 0.29), again as expected. The interaction between group and condition was significant. The subsequent Scheffe post hoc test showed that the D2 value was higher in the ADD group as compared to MCI participants (I–J = 1.978, *p* = 0.039).

#### 3.2.3. Mood and Behavioral Tests (GDS, SAST and NPI)

On the contrary to the majority of the previous studies delivered on this specific research field, but similarly to our previous results, depression, anxiety as well as behavioral symptoms’ D rate measured using the GDS, SAST and NPI, respectively, were not affected due to the second quarantine period, because no significant interaction or main effects were found.

#### 3.2.4. Tests Measuring Short-Term Memory and Visual Perception (RAVLT 1st Trial; Digit forward and backward; RCFT Copy Trial)

High RAVLT trial 1 D’ as well as RCFT copy trial D’ scores indicate decreased performance. According to the RAVLT scores, a significant main effect was found for trial 1. Specifically, D1 (M = −0.361, SE = 0.19) was lower than D2 (M = 0.50, SE = 0.22). In line with this, and interaction between group and condition was found. The subsequent Scheffe post hoc test showed that the mean deterioration difference in D1 in the MCI group was lower than that of the ADD group (I–J = 2.044, *p* = 0.001).

As regards Digit span (forward and backward) test performance, no significant differences were found in people with MCI between 2020 and 2022.

Regarding the ROCFT copy trial, a significant main effect of the group as well as an interaction between group and condition was found. The subsequent Scheffe post hoc test showed that the mean deterioration difference in D2 in the MCI group was higher than that of the ADD group (I–J = 2.112, *p* = 0.001).

#### 3.2.5. Test Measuring Verbal Learning (RAVLT Fifth Trial)

Regarding verbal learning ability (RAVLT fifth trial), there was a significant main effect of deterioration difference and group. Specifically, the verbal learning ability mean deterioration difference in D1 (M = −0.09, SE = 0.24) was lower than that in D2 (M = 0.80, SE = 0.22), whereas people with MCI (M = 0.06, SE = 0.09) had lower D’ than those with ADD (M = 0.64, SE = 0.19). Nevertheless, no significant group–condition interaction was found.

#### 3.2.6. Tests Measuring Visual and Verbal Long-Term Memory (RAVLT Delayed Recall; RCFT Delayed Recall)

As regards the delayed recall condition of the RAVLT, there was a main effect of deterioration rate. The RAVLT mean deterioration rate in D1 (M =−0.54, SE = 0.28) was lower than that in D2 (M = −0.76, SE = 0.27). No significant main effect of the group or group–condition interaction was found.

By means of the RCFT delayed recall, no significant main effect or group–condition interaction was found.

#### 3.2.7. Tests Measuring Verbal Fluency and Executive Functionality

Regarding the results of the phonemic fluency condition, no main effect of the deterioration rate and group or group–condition interaction was found. Repeated measures analysis of variance between D1 (M = 22.53, SE = 12.29) and D2 (M = −44.03, SE = 11.45) in the MCI group regarding their performance in TMT-B showed a significant main effect.

Table 3 shows the indices for main effects and interaction effects (if any) for each test.

### 3.3. The Directed Relationships between Diagnostic Group, Mood, and Cognitive Performance in 2021 and 2022

To establish the potential significant impacts of diagnosis and mood status on cognition after the pandemic period, as well as one year after the official end of quarantine restrictions being imposed (according to the hypothesis of the current study), path analyses were performed separately for the 2021 and 2022 neuropsychological assessments, including only the cognitive variables that were found to decline significantly in terms of deterioration difference. Hence, we entered into the two path models the variables of diagnostic group (two ‘levels’: MCI and ADD), of NPI (as a measure of mood), and those three variables (RAVLT variables) for which significant deterioration differences were found from the aforementioned statistical analyses. The path model for 2021, which was finally confirmed as χ^2^(1, 180) = 1.86, *p* > 0.005, CFI = 0.99, SRMR = 0.02, RMSEA = 0.07 (90%CI: 0.00–0.22), showed that diagnostic group predicted all other variables, as expected, with MCI patients performing better than the ADD group. NPI slightly predicted two of the three scores in RAVLT measures in the expected direction (see Figure 1). The finding that diagnostic group predicted NPI performance, and in this way may indirectly predict cognitive performances, might indicate that the ADD group was more affected by the lockdowns at the mood–behavioral level than MCI patients, and via this pathway, was more affected at the cognitive level too (see Figure 1). However, the higher level of NPI disturbances in ADD patients can simply indicate the severity of their pathology compared to the MCI group.

In order to double check the prototype of relationships between NPI score and RAVLT measures, we ran the same path analysis, using the respective variables for 2022 (see Figure 2). The path model, which was marginally confirmed for 2022 as χ^2^(1, 184) = 3.89, *p* = 0.04, CFI = 0.99, SRMR = 0.03, RMSEA = 0.10 (90% CI: 0.08–0.26), was similar to that of 2021. The model again showed that diagnostic group predicted all other variables, as expected, with MCI patients performing better than the ADD group. NPI slightly predicted two of the three cognitive scores in the expected direction (see Figure 2). Again, the finding that diagnostic group predicted NPI performance, and in this way may indirectly predict cognitive performance, is indicative that people with ADD were probably more affected by the progression of AD at the mood–behavioral level than those with MCI. Therefore, via this pathway, they became more affected at the cognitive level too (see Figure 2).

## 4. Discussion

In line with what was initially hypothesized, the participants of our study did not experience significant cognitive or functional deterioration during the lockdowns, because the deterioration rates between 2020 and 2022 did not differ in the majority of tests administered, except from the RAVLT and TMT B, which was administered only in MCI participants. In accordance, both groups did not demonstrate increased levels of mood or behavioral deterioration after the second lockdown restrictions due to the COVID-19 pandemic.

According to the results, the deterioration rate of participants’ performance in the RAVLT test (subtests which measure short-term memory, verbal learning and long-term memory) was higher between assessments that took place 2021 and 2022 as compared to the same rate of the ones that took place between 2020 and 2021, mainly in people with ADD. This evidence is in accordance with the deterioration rate of our previous results between 2018 and 2020. Therefore, RAVLT score was found to decrease with MCI and ADD progress, which means that this is a quite sensitive means of assessment to identify memory and learning impairments in older adulthood. However, path analyses showed that the RAVLT deterioration rate found in the consecutive 2021–2022 assessments can only be attributed to participants’ condition and consequently their NPI performance. Hence, it can be assumed that the natural progression of MCI as well as ADD is the main reason for participants’ deterioration in the aforementioned test. Trail B has not been included in path models, because it was not administered in participants with ADD.

These results are partially in accordance with those found in our previous study, in which the deterioration rate was increased in RAVLT, FAS and FUCAS tests. The main difference is that after the second wave of the COVID-19 pandemic, only the RAVLT test was found to decline more. Common evidence between our two studies was that tests’ increased deterioration rate was strongly predicted by participants’ condition as well as their NPI performance, rather than any mood or anxiety deteriorations caused by the pandemic.

A large corpus of literature highlights the crucial role of loneliness and isolation in people with mild and major neurodegenerative disorders and mental and cognitive decline. Tam et al. [39] performed an extensive review concerning the experiences and needs of people living with dementia and found increased levels of stress in total as well as levels of loneliness and isolation. Furthermore, they underlined the significance of technology application as a way to increase social connectivity, which is also in line with the study of Li et al. (2022) [40], who found that non-ICT (information and communication technology) users aged more than 80 years with increased levels of isolation and loneliness had increased risk of experiencing cognitive deterioration. To sum up, our findings support the aforementioned results, underlying the significance of digital support during home restrictions, mainly in extensive pandemic situations. An increased sense of loneliness and isolation seems to have a great impact on mood and stress levels in older adults, including those with mild and moderate neurodegenerative disorders, which is in line with previous evidence [41,42,43]. It seems that social isolation negatively affected cognition as well as mental health indexes, which highlights the importance of social interaction in making people living with dementia more active during pandemics. Therefore, dementia services should focus on delivering continuous home support, especially for those who live alone and have other home restrictions. This evidence agrees with previous studies [43,44] which highlighted the importance of maintaining a structured routine including daily support, which is a useful knowledge for professionals who work with people living with ADD. To sum up, data [45] show that those who were socially isolated or consistently remained so during the COVID-19 restrictions self-reported impaired cognition. These results were also confirmed by the study of Farhang et al. (2022) [10], who found poorer cognitive and functional performance in people with MCI due to increased sense of loneliness, as measured by their qualitative data describing their quarantine experience. This study represents significant proof of the crucial role of isolation in cognition rather that the COVID-19 pandemic by itself.

Until now, there have been few data regarding the real effect of pandemic confinement on the cognitive and functional performance of people living with MCI and ADD by means of their clinical and neuropsychological assessment. Data from the study of Kouzuki et al. [46] showed lower cognitive decline in participants with mild cognitive impairment who attended classes for preventing cognitive decline since 2019, focusing on the positive effect of continuous engagement during various kinds of restrictions. In line with this, Kostyál et al. (2021) [47] report that in order to help people living with dementia, it is crucial to provide support to their family members as well as strengthen dementia services. According to their conclusions, it seems that telehealth services need to be enriched beyond the context of COVID-19-pandemic-related social restrictions. Finally, De Pue et al. (2021) [48] found that older adults, the majority of whom lived at home, indicated that their cognitive performance did not change during the COVID-19 pandemic when participating in an online survey. Hence, despite the fact that a vast majority of studies found significant negative impacts of the COVID-19 pandemic and confinement measures, this should be evaluated with caution, due to the multifactorial variables which play a role in this effect.

On the contrary, data from various studies such as that by Ismail et al. (2021) [15] demonstrated statistically significant impairment in 36 people with MCI and dementia who had several follow up measurements before the pandemic and were also tested after the pandemic. Nevertheless, they used only MMSE tests without taking into account the possible impact of disease progression which could also be assumed as a contributing factor of their evidence. Moreover, no data about the role of mood disturbances are reported. Of foremost importance are the results of Tondo et al. (2021) [49], who found cognitive deterioration in people living with dementia, irrespective of etiology, just after the imposition of the restrictions related to the COVID-19 in comparison to the other groups who had been evaluated in previous years; the 2019 group and 2018 group. A significant main effect was also found in the 2020 group by means of their MMSE performance during the 2019–2020 period. However, in their study, they used only the MMSE score, in contrast with our study, in which various cognitive, functional, mood and behavioral tests were used. Another significant issue is that no regression models comparing the possible impact of mood and behavioral disorders due to the pandemic on participants’ cognitive performance were applied, whereas no relevant path models were implemented to exclude the significant effect of dementia progression in MMSE scores. Another difference is that in our study, older adults with MCI were also recruited. Additionally, Amieva et al. (2022) [50] reported accelerated cognitive decline in older adults who participated in epidemiological studies 15 years before the pandemic, who were also tested during the pandemic. However, they used MMSE test performance before the pandemic, whereas during the pandemic, they carried out a telephone interview for cognitive status, as both tests share 11 items. In summary, in the aforementioned studies, they neither used an entire neuropsychological battery in people with MCI and ADD nor clinically validated tools for the assessment of BPSD symptoms. Therefore, no clear comparisons in relation to our study can be made.

A significant novelty of our studies is the use of deterioration rate across years. Through the calculation of the deterioration rate and not just participants’ raw data, the significant role of diagnosis and the subsequent cognitive and functional decline could be better identified. Moreover, the configural path models applied further described the significant impact of diagnosis, mood and behavioral disturbances on various aspects of cognition and everyday functioning before, during and after lockdowns.

To sum up, one can assume that the lower performance found in working memory and learning ability, as measured using the RAVLT test, as well as cognitive flexibility, as measured using the TMT B, observed in people with MCI can be assumed to be due to NPI decline and disease progression. In specific, according to Figure 1 and Figure 2, we found that learning, verbal long-term memory, short-term memory—which deteriorated faster between the 2021 and 2022 assessments compared to those between 2020 and 2021—were predicted by the NPI test. Therefore, NPI was found to predict deterioration irrespective of the quarantine being imposed. Finally, despite the fact that TMT-B seems to worsen in the MCI population, it was not inserted into the path model, because it was not administered in the ADD group.

## 5. Limitations

Firstly, in the current study, we recruited a relatively small number of participants with ADD, because they skipped the follow up tests despite the fact that they had been informed of them by our research team. Secondly, we did not conduct any separate analyses for the small minority who had not been diagnosed with COVID-19s in our sample. Finally, no self-report measures were administered by means of participants’ sense of isolation, or possible negative feelings or supportive variables during the second wave of pandemic. Furthermore, the generalizability of our study is a significant issue because the non-pharmacologic interventions provided at our day centers may not be applicable elsewhere.

## 6. Conclusions

Future studies should shed light on the effect of lockdowns, investigating the longitudinal impact of the COVID-19 pandemic on cognition, mood and daily functionality in older adults. According to our two studies, the increased deterioration rates found in the tests measuring cognition and daily functionality (mainly after the first wave of the pandemic) were mainly attributed to NPI and disease progression. A main explanation of the above findings is the fact that these participants were active during the pandemic in attending cognitive training in a digital mode, whereas they had continuous medical and psychological support during this period. To our knowledge, there are no similar studies to identify the possible protective role of cognitive training in cognition, functionality, mood and behavior in those with MCI and ADD during extensive home restrictions. Furthermore, the authors underline the possibly protective role of ICT in older adults who live alone. Hence, the role of social isolation and an increased sense of loneliness combined with a lack of practical and psychological support should be further assessed to enrich our knowledge regarding future restrictive situations.

## Figures and Tables

**Figure 1 brainsci-13-01044-f001:**
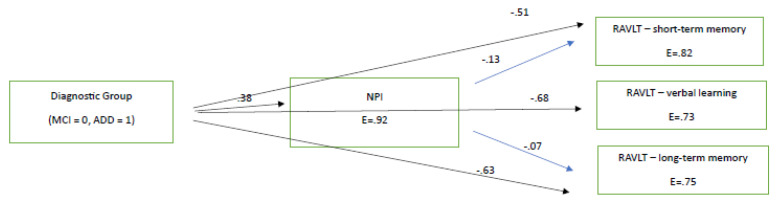
The directional relationships between diagnostic group and neuropsychiatric disturbances and cognitive measures in 2021 (immediately after the lockdown). All paths are significant at *p* < 0.05. E = measurement error. All RAVLT variables are significantly related between each other at *p* < 0.05.

**Figure 2 brainsci-13-01044-f002:**
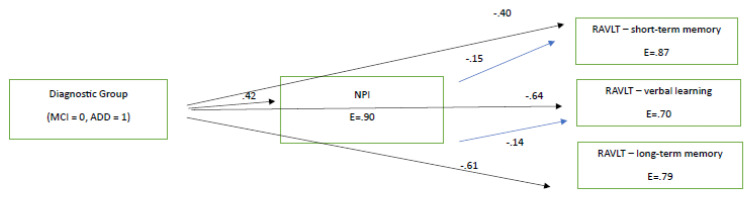
The directional relationships between diagnostic group and neuropsychiatric disturbances and cognitive measures in 2021 (immediately after the lockdown). All paths are significant at *p* < 0.05. E = measurement error. All RAVLT variables are significantly related between each other at *p* < 0.05.

**Table 1 brainsci-13-01044-t001:** Demographic distribution in participants with MCI and ADD.

Demographic Variables	MCI (n = 175)	ADD (n = 35)
Gender		
*Men (n)*	59	17
*Women (n)*	116	18
Age (years) M (SD)	71.59 (6.9)	77.94 (6.4)
Education (years) M (SD)	12.65 (3.7)	9.83 (5.1)

**Table 2 brainsci-13-01044-t002:** Mean and standard deviation of the neuropsychological tests and mood–behavioral tests in MCI and ADD participants.

Assessment Tools	Group	2021	2022
		Mean (SD)	Mean (SD)
MMSE	MCI	27.82 (2.1)	27.63 (2.3)
	ADD	19.00 (5.0)	16.91 (6.8)
MoCA	MCI	24.89 (3.8)	24.85 (3.6)
	-	-	
FUCAS	MCI	44.04 (3.7)	44.35 (4.6)
	ADD	58.59 (13.9)	62.61 (16.27)
FRSSD	MCI	3.24 (2.53)	3.33 (3.08)
	ADD	10.20 (6.08)	13.17 (8.32)
NPI	MCI	2.47 (3.8)	2.36 (3.4)
	ADD	8.79 (12.1)	9.05 (12.05)
SAST	MCI	15.99 (4.4)	16.21 (4.3)
	ADD	13.19 (3.6)	13.17 (3.1)
GDS	MCI	2.46 (2.8)	2.1 (2.6)
	ADD	2.44 (2.6)	2.90 (3.58)
RAVLT1	MCI	6.33 (2.1)	5.44 (2.1)
*Short term memory*	ADD	2.81 (1.5)	2.61 (2.1)
RAVLT 5	MCI	11.47 (2.7)	10.65 (2.6)
*Verbal learning*	ADD	4.78 (2.9)	3.97 (2.7)
RAVLT	MCI	9.02 (3.6)	7.82 (3.6)
*Delayed Recall*	ADD	1.34 (2.4)	0.88 (2.2)
RCFT	MCI	30.71 (5.05)	30.23 (4.6)
*Copy*	ADD	17.70 (11.02)	18.63 (11.09)
RCFT	MCI	17.70 (7.1)	17.41 (7.7)
*Delayed Recall*	ADD	3.2 (5.16)	1.7 (2.2)
Phonemic fluency	MCI	12.97 (3.7)	12.84 (3.8)
	ADD	7.29 (3.30)	6.33 (3.9)
TMT B	MCI	186.68 (119.76)	215.34 (145.77)
	ADD	-	-

Abbreviations: Mild cognitive impairment: MCI; Alzheimer’s disease dementia: ADD; standard deviation: SD; Mini Mental State Examination: MMSE; Montreal Cognitive Assessment: MoCA; Functional Cognitive Assessment Scale: FUCAS; Neuropsychiatric Inventory: NPI; Short Anxiety Screening Test: SAST; Rey Auditory Verbal Learning Test (RAVLT); Rey Complex Figure Test: RCFT; Trail Making Test B: TMT B.

**Table 3 brainsci-13-01044-t003:** Results of the 2 × 2 mixed measures ANOVA in MCI and ADD groups across the deterioration rate levels (D1 and D2).

	Deterioration Rate (D)	Group	Group by D Interaction
	*F* *p* η^2^	*F* *p* η^2^	*F* *p* η^2^
MMSE	0.21 0.642 0.001	34.3 <0.001 ** 0.14	0.77 0.379 0.004
MoCA	0.537 0.465 0.003	------	------
FUCAS	3.32 0.070 0.017	42.21 <0.001 ** 0.17	5.07 0.025 * 0.026
FRSSD	2.63 0.106 0.013	36.65 <0.001 * 0.15	4.29 0.039 * 0.022
NPI	2.84 0.093 0.016	5.46 0.021 0.029	1.61 0.206 0.009
SAST	2.03 0.155 0.010	0.055 0.815 0.000	0.008 0.929 0.000
RAVLT	5.57 0.019 * 0.031	3.32 0.070 0.019	7.29 0.008 * 0.040
*First trial*			
RCFT	3.58 0.055 0.013	31.21 <0.001 * 0.146	11.70 0.001 ** 0.060
*Copy*			
RAVLT	4.93 0.028 * 0.028	6.83 0.010 * 0.038	1.90 0.169 0.01
*5th trial*			
RAVLT	7.31 0.008 * 0.040	0.45 0.501 0.003	3.47 0.064 0.02
*Delayed*			
RCFT	4.44 0.059 0.009	1.55 0.175 0.010	2.33 0.128 0.013
*Delayed*			
Verbal	0.160 0.689 0.001	0.27 0.601 0.001	0.179 0.673 0.001
Fluency			
TMT B	5.09 0.025 * 0.032	------	------

Abbreviations: Mild cognitive impairment: MCI; Alzheimer’s disease dementia: ADD; standard deviation: SD; Mini Mental State Examination: MMSE; Montreal Cognitive Assessment: MoCA; Functional Cognitive Assessment Scale: FUCAS; Neuropsychiatric Inventory: NPI; Short Anxiety Screening Test: SAST; Rey Auditory Verbal Learning Test (RAVLT); Rey Complex Figure Test: RCFT; Trail Making Test B: TMT B. * *p* < 0.05; ** *p* < 0.01.

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
