# Peer review of "Cognition and Functionality Were Not Affected Due to the COVID-19 Pandemic in People with Mild Cognitive Impairment and AD Dementia Attending Digital Non-Pharmacologic Interventions"

_brainsci, 2023, doi:10.3390/brainsci13071044_

Round 1

Reviewer 1 Report

Comments and Suggestions for Authors

This paper reports on a two-centre study on the effects of the second quarantine during the COVID-19 pandemic in Greece on cognition, mood and behavior of patients with MCI or ADD attending non-pharmacologic interventions in their centres. They found that there was no significant impact except on verbal memory, and felt any changes were due to the underlying illness and not the pandemic restrictions – this is largely contrary to other studies.
The authors may wish to attend to these more pressing issues, many due to the lack of adequate command of the English language, which made reading the paper challenging:
1. Title – reword as “Cognition, mood and behaviour were not affected among people with Mild Cognitive Impairment or AD Dementia attending non-pharmacologic interventions at Day Centres’
2. Abstract – Methods - to mention that two Day Centres were involved. Results – to provide patient demographics. The word ‘eliminate’ may not be the most appropriate. Conclusion – 2nd sentence should be part of the Results; first and last sentence need to be rationalised    
3. Throughout the text - excessive and inappropriate use of linking words such as interestingly, additionally, also moreover, in particular, therefore, in specific, specifically, characteristically, in addition, although, while it is worth noting, etc – these examples are taken only from the Introduction, they are repeated in other parts of the paper
4. Introduction – the hypothesis far too long and wordy, and should add about the setting of the study ie attending non-pharmacologic interventions at Day Centres
5. Methods - 2.4 Data Analysis – please re-look at how D3 and D4 are defined here, they seem inconsistent and confusing
6. Fig 2 – is the right side missing?
7. Discussion – para 1 should be removed and merged into the Introduction.
8. Conclusions - sentence 4 should be in the Discussion
9. Limitations - should appear before Conclusions? More limitations should be about get generalisablity of the study as the non-pharmacologic interventions provided at these 2 centres performed on the German population may not be applicable elsewhere

Comments on the Quality of English Language

Needs attention

Author Response

R1C1: Title – reword as “Cognition, mood and behaviour were not affected among people with Mild Cognitive Impairment or AD Dementia attending non-pharmacologic interventions at Day Centres’

R1R1: The Title has now changed.

R1C2: Abstract – Methods - to mention that two Day Centres were involved. Results – to provide patient demographics. The word ‘eliminate’ may not be the most appropriate. Conclusion – 2nd sentence should be part of the Results; first and last sentence need to be rationalized

R1R2: Abstract section has now changed .

R1C3: Throughout the text - excessive and inappropriate use of linking words such as interestingly, additionally, also moreover, in particular, therefore, in specific, specifically, characteristically, in addition, although, while it is worth noting, etc – these examples are taken only from the Introduction, they are repeated in other parts of the paper

R1R3: Linking words has been significantly reduced throughout the paper

R1C4: Introduction – the hypothesis far too long and wordy, and should add about the setting of the study ie attending non-pharmacologic interventions at Day Centres

R1R4: The hypothesis has been reduced. As regards the description of the non-pharmacologic interventions, they are described in more details in our previous paper (Tsatali et al., 2021)

R1C5: Methods - 2.4 Data Analysis – please re-look at how D3 and D4 are defined here, they seem inconsistent and confusing

R1R5: This point has been now corrected.

R1C5: Fig 2 – is the right side missing?

R1R5: I attach the entire Fig 2 in the resubmission

R1C6: Discussion – para 1 should be removed and merged into the Introduction.

R1R6: Done

R1C7: Conclusions - sentence 4 should be in the Discussion

R1R7: Done

R1C8: Limitations - should appear before Conclusions? More limitations should be about get generalizability of the study as the non-pharmacologic interventions provided at these 2 centers performed on the German population may not be applicable elsewhere

R1R8: Done

Reviewer 2 Report

Comments and Suggestions for Authors

Comments to the Authors

Ø  The heading can be made concise without omitting of much needed information.

Ø  In the second last paragraph of the Introduction, it is non-pharmaceutical or non-pharmacological?

Ø  If the people were already affected by MCI and AD, how would it be logical to further assess their cognitive impairment because of Covid-19.

Ø  The data other than Greece also could have been included in or at least cited.

Ø  The references can be improved and updated, like the recently published article related to impact of Covid-19 on brain and psychological health  http://dx.doi.org/10.2174/1872208316666220617110402 etc. can be cited.

Ø  As mentioned in the Aim, “second wave of pandemic” the waves were at different times in different countries. Please specify in other parts of the manuscript too.

Ø  Have the participants recovered from Covid-19? Were they vaccinated? Which precautions were taken to avoid infection were taken or not during assessment?

Comments on the Quality of English Language

Author Response

R2C1: The heading can be made concise without omitting of much needed information.

R2R1: Heading has changed.

R2C2: In the second last paragraph of the Introduction, it is non-pharmaceutical or non-pharmacological?

R2R2: Non pharmacological

R2C3: If the people were already affected by MCI and AD, how would it be logical to further assess their cognitive impairment because of Covid-19.

R2R3: Because a large corpus of literature shows that those with MCI and AD were affected due to the Covid 19 regarding their cognitive mood and behavioral level. Additionally, people with MCI and AD do follow up assessments every one or two years.

R2C4: The data other than Greece also could have been included in or at least cited.

R2R4: We included also studies from populations other than Greek.

R2C5: The references can be improved and updated, like the recently published article related to impact of Covid-19 on brain and psychological health  http://dx.doi.org/10.2174/1872208316666220617110402 etc. can be cited.

R2R5: They are not in line with the one suggested from Brain Sciences journal

R2C6 As mentioned in the Aim, “second wave of pandemic” the waves were at different times in different countries. Please specify in other parts of the manuscript too.

R2R6: We have identified the exact pandemic waves in Greece (see in Methods section)

R2C7: Have the participants recovered from Covid-19? Were they vaccinated? Which precautions were taken to avoid infection were taken or not during assessment?

R2R7: People did not have Covid 19 during the assessment. They provided negative rapid tests before the assessment. To ensure precautions, all participants showed negative Covid 19 tests before coming and they also weared masks, as well as the personnel

Round 2

Reviewer 1 Report

Comments and Suggestions for Authors

This paper is a revised submission of a report on a two-centre study on the effects of the second quarantine during the COVID-19 pandemic in Greece on cognition, mood and behavior of patients with MCI or ADD attending non-pharmacologic interventions in their centres.

As there is no ’marked-up’ version, I am unable to see exactly where the changes were made.

Still, the paper appears improved. There remain some minor issues:

1.       Title – it is really long now – the Editor may wish to advise a shorter version

2.       Abstract – Methods – I suggest that ‘ All participants underwent three yearly follow up assessments from May 2020 to May 2022, participated in cognitive training interventions (through digital online means) during the aforementioned period. All of them were recruited from the two Day Centers of the Greek Association of Alzheimer Disease and Related Disorders (GAADRD)’ be re-worded as ‘Participants were recruited from the two Day Centers of the Greek Association of Alzheimer Disease and Related Disorders (GAADRD). They underwent three yearly follow up assessments from May 2020 to May 2022, participated in cognitive training interventions (through digital online means) during the aforementioned period’. Results – to add data on gender.

3.       Fig 2 – the right side is still missing….

Author Response

As there is no ’marked-up’ version, I am unable to see exactly where the changes were made.

Rev1 is right. Actually we did extensive changes throughout the manuscript , so if we would do track changes, the manuscript would be non readable

1.Title – it is really long now – the Editor may wish to advise a shorter version

Any ammendements by the Editor are welcome

 2.    Abstract – Methods – I suggest that ‘ All participants underwent three yearly follow up assessments from May 2020 to May 2022, participated in cognitive training interventions (through digital online means) during the aforementioned period. All of them were recruited from the two Day Centers of the Greek Association of Alzheimer Disease and Related Disorders (GAADRD)’ be re-worded as ‘Participants were recruited from the two Day Centers of the Greek Association of Alzheimer Disease and Related Disorders (GAADRD). They underwent three yearly follow up assessments from May 2020 to May 2022, participated in cognitive training interventions (through digital online means) during the aforementioned period’. Results – to add data on gender.

Done

3. Fig 2 – the right side is still missing….

Ι have attached it in a separate email

Reviewer 2 Report

Comments and Suggestions for Authors

The authors have responded to the comments and completed the required additions or deletions.

Comments on the Quality of English Language

The manuscript can be improved, however, other reviewers and Editor's comments must be consider

Author Response

We want to thank Rev 2 for his/her comments!Minor editing of English language has been made.